Immediate effects of EVA midsole resilience and upper shoe structure on running biomechanics: a machine learning approach

Onodera Andrea N. 1 2
Gavião Neto Wilson P. 3
Roveri Maria Isabel 1
Oliveira Wagner R. 2
Sacco Isabel CN icnsacco@usp.br 1
1 Physical Therapy, Speech and Occupational Therapy Department, University of São Paulo, School of Medicine , São Paulo , Brazil
2 Dass Nordeste Calçados e Artigos Esportivos Inc , Ivoti , Rio Grande do Sul , Brazil
3 School of Engeneering & IT, Centro Universitário Ritter dos Reis , Porto Alegre , Rio Grande do Sul , Brazil
Zhang Yu-Dong
Electronic publication date: 2017 Feb 28
Publication date: 2017
Volume: 5
Electronic Location ID: e3026
Received 2016 Oct 26; Accepted 2017 Jan 25
Copyright: ©2017 Onodera et al.
Copyright year: 2017
Copyright holder: Onodera et al.
License: This is an open access article distributed under the terms of the Creative Commons Attribution License, which permits unrestricted use, distribution, reproduction and adaptation in any medium and for any purpose provided that it is properly attributed. For attribution, the original author(s), title, publication source (PeerJ) and either DOI or URL of the article must be cited.
License URL: https://creativecommons.org/licenses/by/4.0/

Keywords: Shoes, Running, Kinetics, Biomechanics, Neural networks, Kinematics, Machine learning

Funding: Coordination of Improvement of Higher Education Personnel (CAPES) National Council for Scientific and Technological Development (CNPq) I.C.N. Sacco Process: 305606/2014-0 Agency Coordination of Improvement of Higher Education Personnel (CAPES) provided Roveri’s scholarship. National Council for Scientific and Technological Development (CNPq) funded I.C.N. Sacco (Process: 305606/2014-0). The funders had no role in study design, data collection and analysis, decision to publish, or preparation of the manuscript.

==============================
Background

Resilience of midsole material and the upper structure of the shoe are conceptual characteristics that can interfere in running biomechanics patterns. Artificial intelligence techniques can capture features from the entire waveform, adding new perspective for biomechanical analysis. This study tested the influence of shoe midsole resilience and upper structure on running kinematics and kinetics of non-professional runners by using feature selection, information gain, and artificial neural network analysis.

Methods

Twenty-seven experienced male runners (63 ± 44 km/week run) ran in four-shoe design that combined two resilience-cushioning materials (low and high) and two uppers (minimalist and structured). Kinematic data was acquired by six infrared cameras at 300 Hz, and ground reaction forces were acquired by two force plates at 1,200 Hz. We conducted a Machine Learning analysis to identify features from the complete kinematic and kinetic time series and from 42 discrete variables that had better discriminate the four shoes studied. For that analysis, we built an input data matrix of dimensions 1,080 (10 trials × 4 shoes × 27 subjects) × 1,254 (3 joints × 3 planes of movement × 101 data points + 3 vectors forces × 101 data points + 42 discrete calculated kinetic and kinematic features).

Results

The applied feature selection by information gain and artificial neural networks successfully differentiated the two resilience materials using 200(16%) biomechanical variables with an accuracy of 84.8% by detecting alterations of running biomechanics, and the two upper structures with an accuracy of 93.9%.

Discussion

The discrimination of midsole resilience resulted in lower accuracy levels than did the discrimination of the shoe uppers. In both cases, the ground reaction forces were among the 25 most relevant features. The resilience of the cushioning material caused significant effects on initial heel impact, while the effects of different uppers were distributed along the stance phase of running. Biomechanical changes due to shoe midsole resilience seemed to be subject-dependent, while those due to upper structure seemed to be subject-independent.

Introduction

Sports shoes have many roles in running; among them, providing adequate impact-force absorption (Clarke, Frederick & Cooper, 1983; Hennig, 2011), stability for foot/ankle movements (Cheung, Wong & Ng, 2011) and comfort (Nigg, 2010). These roles have been the most studied in running and shoe biomechanics so far. Running shoes are basically constituted by upper, midsole and sole. Among different possible combinations of these three elements in the shoe construction, the upper is definitely the most prone to variations in its construction, such as color, model design, added elements and materials, and certainly, the last two factors will have a particular influence on running biomechanics. Runners select a comfortable running shoe using their own comfort criteria (Nigg et al., 2015) and, because the shoe upper maintains a large contact area with the foot, it would have a stronger influence over fit and comfort, which in turn would impact in runner’s kinematic and kinetic strategies during practice and competitions. It has been demonstrated that a firmer foot contact with a shoe resulted in lower loading rates due to a better coupling of foot-footwear, which optimizes the use of the midsole impact absorption technology by favoring a better foot positioning inside the shoe (Hagen & Hennig, 2009). To investigate the isolated influence of upper types in running biomechanics would help runners to have a more comprehensive and efficient approach in the shoe construction process, as well as in the choice of the running shoe by runners.

Nevertheless, the most manipulated and studied shoe part in biomechanics is still the midsole (Maclean, Davis & Hamill, 2009; Milani, Hennig & Lafortune, 1997; Nigg et al., 2003; Worobets et al., 2014). The majority of shoe companies invests a large amount of time, effort and money on development of damping materials technologies, such as gels, air, and springs for supposedly improving sports performance. Ethylene-vinyl acetate (EVA) is a copolymer of ethylene and vinyl acetate highly elastic sintered to form a porous material similar to rubber, yet with excellent toughness. Its porous elastomeric characteristic is much more flexible as low-density polyethylene, commonly used in shoes construction, and because of its properties of resistance, flexibility, temperature toughness, it has been one of the most used copolymer in the shoe midsole construction (Verdejo & Mills, 2004; Wang, Hong & Li, 2012).

The midsole hardness is the most explored physical characteristic of the midsole in biomechanical studies (Clarke, Frederick & Cooper, 1983; Hennig, Valiant & Liu, 1996; Kersting & Brüggemann, 2006; Maclean, Davis & Hamill, 2009; Milani, Hennig & Lafortune, 1997; Nigg et al., 2012). Running with hard shoes resulted in same peak magnitude vertical GRF (ground reaction force) as running with soft ones (Clarke, Frederick & Cooper, 1983; Kersting & Brüggemann, 2006; Nigg et al., 1987) and faster time to achieve the first peak (Clarke, Frederick & Cooper, 1983). Therefore, midsole hardness affected the loading rate but not in a proportional rate (Milani, Hennig & Lafortune, 1997). Most of running kinematic changes due to the midsole hardness occurs at the ankle joint (Clarke, Frederick & Cooper, 1983; Hardin, Van den Bogert & Hamill, 2004; Maclean, Davis & Hamill, 2009) and some authors state that these different midsole hardness lead to different impact perception by runners (Hennig, Valiant & Liu, 1996; Milani, Hennig & Lafortune, 1997), which in turn causes distinct alteration on running mechanics (Kersting & Brüggemann, 2006; Nigg et al., 2012) and may mislead the real impact damping by shoe midsoles (Hennig, Valiant & Liu, 1996; Milani, Hennig & Lafortune, 1997).

Apart from midsole hardness, resilience is also an important mechanical property of midsole that has been seldom studied (Sinclair et al., 2014; Sinclair et al., 2016; Worobets et al., 2014). It represents the energy restored by the cushioning material after an applied force ceases. Managing resilience while maintaining the hardness of a polymeric foam is possible by adding different kinds of compounds to its formula. EVA added to elastomers could have ideal softness and high resilience characteristics, would have a full-recovery capacity for the next foot step after a heel strike, while a less resilient (more viscous) material would have the capacity of attenuating more energies at initial loading cycles, easily achieving compression flattening after some cycles. It is expected that different resilience materials would mainly reflect different initial impact forces, because more resilient materials will quickly restore the cushioning property while less resilient materials will take a little longer to restore the cushioning property (Sun et al., 2008). Sinclair et al. (2014) have shown that running with shoes with energy return component resulted in greater tibial acceleration peak, calcaneous eversion and internal tibia rotation compared to conventional running shoes. In a later study, they have shown lower oxygen consumption and respiratory exchange ratio with more resilient model of shoes (Sinclair et al., 2016). Worobets et al. (2014) manipulated only the midsole materials, maintaining the upper structure, and also reported lower oxygen consumption when running with a more compliant/resilient midsole condition. However, the isolated effect of resilience changing in shoe midsole is still unknown in running biomechanics.

The majority of biomechanics studies vary the shoe model as a whole to investigate the effects of various structural shoe properties and elements while running (Azevedo et al., 2012; Braunstein et al., 2010; Dixon, 2008; McNair & Marshall, 1994). Such an approach deeply interferes with an appropriate differentiation and interpretation of which shoe characteristics most influence the kinetic and kinematic changes during running. The novelty of this study proposal was to manipulate selectively the upper structure and the cushioning material resilience and to investigate the effects of this manipulation in the biomechanics of running. Identifying more precisely which shoe characteristics really matter for impact attenuation and lower limb kinematic adaptation would help runners to choose more wisely the running shoes regardless the brand or model and direct further running training regimes based on that choice.

Nonetheless, individual’s mechanical and neuromuscular adaptations to changes in shoes are influenced by mechanical, neurophysiological, anatomical and even psychological factors and, therefore, is likely to observed different individuals using different strategies in response to changes in running shoes (Kersting & Brüggemann, 2006; Nigg et al., 2003). Thereby, one may conclude that regardless the type of shoe modification, the biomechanical responses observed may be subject-dependent.

We proposed to identify the relevant biomechanical features which are most affected by different shoes conditions during running, two midsole resiliencies and two upper structures. We adopted an approach based on machine learning (ML), which has been used in the literature to identify crucial features and relevant patterns for classifying and predicting locomotor patterns as being a result of a given health condition (Hoerzer et al., 2015; Muniz et al., 2010b; Schöllhorn et al., 2002), but it has never been used before to identify effects of different running shoes on biomechanics. In contrast to the related literature, our ML strategy is entirely supervised, and it consisted of using Information Gain (IG) to select important features and Artificial Neural Networks (ANN) to classify the different shoe resiliencies and upper types. Our assumptions were: (h1) low versus high resilient cushioning effects on running kinematics and ground reaction forces are classifiable by using a ML approach; (h2) structured versus minimalist upper effects on running kinematics and ground reaction forces are classifiable by using a ML approach; and (h3) that there are biomechanical changes due to shoe midsole resilience or upper structure that are subject-independent.

Methods

Subjects

Twenty-seven experienced non-professional male runners (36.0 ± 7.3 years old, 1.72 ± 0.05 m, 73.9 ± 6.2 kg, 62.9 ± 43.8 km/week run, 7.5 ± 7.1 years of practice) with a rearfoot strike pattern and with no experience in minimalist shoe participated in this study. All runners were free of injury or musculoskeletal pain according to the definition of Macera et al. (1989); had no major foot or ankle postural alterations or deformities, excessive static pronation or supination of the foot and ankle complex according to the Foot Posture Index (Redmond, Crosbie & Ouvrier, 2006); and did not present leg length discrepancy greater than 1 cm. All athletes that participated in the Porto Alegre’s Marathon in 2012 were invited by electronic media, and 158 athletes answered positively to participate in the study; however, only 27 effectively matched the eligibility criteria and came to the lab to be evaluated. Then, a telephone call was made to select runners that fit to the inclusion criteria and scheduled the biomechanical evaluation in the laboratory. All subjects agreed to participate in the study approved by the Ethics Committee of the School of Medicine of the University of Sao Paulo (Ethical Application CEP-FMUSP: protocol #054/14) and signed a written consent form.

Tested running shoes

Four running shoes were especially developed by a local sportive shoe manufacturer. The final masses of the 4 constructions were equivalent to avoid negative effects due to mass differences (Frederick, 1984). All shoes were constructed using the same last, the same design of upper pieces, midsole geometry, and outsole. The hardness of cushioning materials was fixed at 40 Asker C, measured by a durometer (GoTechAskerC, Taichung, Taiwan). The two shoe uppers had the same design and shape, but the different structure and materials:

(1) SU—structured upper: 15 mm of soft foam in the heel collar and tongue, hard heel cup involving the medial, lateral and posterior parts of the heel, synthetic pieces sewed in the vamp and doubled fabric over the whole shoe (Fig. 1A).

(2) MU—minimalist upper: light-weight mesh, tongue without foam, without heel cup, and almost all pieces of the upper were connected by means of heat fusion (Fig. 1B).

Both cushioning materials were made of ethylene-vinyl acetate (EVA); were inserted in the same rearfoot area within the midsole; and had an oval shape of 10 mm thickness, 50 mm width, and 70 mm length. Resilience was assessed by vertical resiliometer (GoTech GT7042-V1, Taichung, Taiwan). The midsoles were: (1) LR—low resilience—5% of resilience (±3%) (Fig. 1C), and (2) HR—high resilience—55% of resilience (±3%) (Fig. 1D).

Figure 1 Image of shoes prototypes used in the experiment.

Illustration of testing shoes. (A) Structured Upper condition (SU). (B) Minimalist Upper condition (MU). (C) Low Resilience cushioning material condition (LR). (D) High Resilience cushioning material condition (HR).

The first tested condition (condition 1—upper SU and cushioning material LR) was the same for all runners; the other three testing conditions were randomized for each subject using simple draw. The other three conditions were: condition 2—upper SU and cushioning material HR condition 3—upper MU and cushioning material LR, and condition 4—upper MU and cushioning material HR. The subjects were asked to lace their shoes tightly and comfortably, in the same way they typically lace during their running practice.

Experimental protocol

Kinematic data were acquired by six infrared cameras (VICON T-40, Oxford, UK). Sixteen passive-reflexive markers (14 mm diameter) were fixed on both lower limbs (two anterior superior iliac spines; two posterior iliac spines; two lateral epicondyles of the knees; two markers over the lower lateral 1/3 surface of the thighs; two lateral malleolus; two markers over the lower 1/3 of the shank; two second metatarsal heads; two posterior surface of calcaneous at the same height above the plantar surface of the foot as the toes markers) according to Plugń Gait marker set (Kadaba, Ramakrishnan & Wootten, 1990). The two foot markers were fixed on the shoes (second metatarsal heads and calcaneous) after deep palpation of bone prominences. The laboratory coordinate system was established at one corner of one force plate, and all initial calculations were based on this global coordinate system. In Nexus software (Vicon Nexus 1.7, Oxford, UK), each data sample from each lower limb segment (foot, shank, and thigh) was modeled as a rigid body with a local coordinate system that coincided with anatomical axes. Translations and rotations of each segment were reported relative to neutral positions defined during the static standing trial.

The program calculates the joint angles by means of a decomposition matrix based on Cardan sequences and six degrees of freedom model. The decomposition matrix describes the relationship between two local coordinate systems, one for each segment between which the relative angle is determined. The joint kinematics was considered as the movement of the distal segment in relation to the proximal; e.g., for determining the knee angle, the thigh was the proximal segment and the shank the distal one. The movements occur around 3 different axes which describe two definition of movement each: flexion/extension, abduction/adduction, and internal/external rotation (Hamill, Selbie & Kepple, 2014).

Ground reaction forces were acquired at 1200 Hz by two force plates (AMTI BP600600, Watertown, USA) embedded in the center of a 25 m walkway. Acquisitions of kinematic and force data were synchronized by a 64 multichannel Vicon MX Giganet Lab and A/D converter.

Running velocity was kept between 9.5 and 10.5 km/h (mean 10.1 ± 0.5 km/h), monitored by 2 photoelectrical sensors (Tecsistel Speed View, Novo Hamburgo, Brazil). Ten trials per subject for each shoe condition were collected, resulting in 40 trials on the dominant limb. The limb dominance was defined as the leg used to kick a soccer ball (Greenberger & Paterno, 1995).

Biomechanical data analysis

In accordance with many studies in biomechanics (Muniz et al., 2010b; Nigg et al., 2012), we adopted a Butterworth filter (implemented using the original code from MATLAB) to minimize noise, and the marker coordinates were filtered using a 12 Hz zero-lag fourth-order low-pass Butterworth filter. Force data was filtered with a 300 Hz zero-lag fourth-order low-pass Butterworth filter also implemented in a MATLAB code. The angular and force data from initial contact to take-off were normalized in stance time (interpolated 0–100%) and in magnitude by the body weight.

The 30 discrete kinematic features analyzed were: peak angles (degrees), angles at the beginning of stance phase (degrees), instant of peak angle (seconds), range of motion from the beginning of stance phase to peak angle (degrees), and final angle of stance phase (degrees); for ankle, knee and hip joints; for sagittal and frontal planes of movement (5 × 3 × 2).

For vertical and antero-posterior forces, the magnitude of first vertical force peak (1VFP) (body weight), time of 1VFP (milliseconds), magnitude of second vertical force peak (2VFP)(body weight), time of 2VFP (milliseconds), loading rate (slope of 20%–80% of 1VFP) (body weight/second), time of minimal vertical force in midstance (milliseconds), propulsion rate (slope of curve between minimal vertical force in midstance and the 2VFP) (body weight/second), minimum breaking antero-posterior force (body weight), breaking antero-posterior impulse (body weight*second), median frequency of 1VFP (Hz), time of stance phase (milliseconds), and decay rate (slope of curve from 2VFP to the end of stance phase) (body weight/ second) were calculated.

The whole interpolated time-series of all three planes of motion (sagittal, frontal and transversal) and forces (vertical, antero-posterior and medio-lateral) were also analyzed. Usually, cross-sectional studies that investigate shoe effects in running biomechanics involve high-dimensional and redundant datasets (Maurer et al., 2012; Nigg et al., 2012), and feature selection techniques have been used to help identifying the biomechanical parameters that is most influenced by shoe characteristics (Hoerzer et al., 2015; Maurer et al., 2012; Nigg et al., 2012). This was the main reason why we chose to include both discrete and whole time-series points in the analysis. As explained in the next sections, we adopted an approach based on ANN, which can receive large numbers of data simultaneously and the pieces of data do not have to be isolated from each other (Barton & Lees, 1997).

Machine learning approach

To assess the effects of shoe interventions, many studies have generated high-dimensional and redundant datasets (Maurer et al., 2012; Nigg et al., 2012), which impose challenges for understanding an underlying phenomenon of interest (Guyon et al., 2006; Yu & Liu, 2003). To overcome these challenges, Machine Learning (ML) techniques have been adopted to find patterns on biomechanical data (Begg & Kamruzzaman, 2005; Hoerzer et al., 2015; Maurer et al., 2012; Muniz et al., 2010b; Nigg et al., 2012). ML aims to learn from data. In a typical classification scenario, we have a categorical outcome (like low vs high shoes resilience) that we wish to predict or classify based on a set of features or variables (like ground reaction forces). On the basis of a training set of data, we observe the outcome and feature measurements for a set of instances (like the subject’s trials) (Hastie, Tibshirani & Friedman, 2001). Using this data, we build a classification model, which will enable us to classify the outcome for new unseen instances. A good model is one that accurately classify such an outcome. This scenario characterizes a supervised learning problem since each observed instance involves the outcome variable (i.e., the desired output value) to guide the learning process.

A typical supervised classification approach consists of two parts: (i) variable or feature selection and (ii) classification. Information Gain (IG) and Artificial Neural Networks (ANN) are ML techniques that have been successfully used for feature selection and classification in many areas. Whereas ANN have been used to classify patterns in biomechanics studies (Hastie, Tibshirani & Friedman, 2001; Muniz et al., 2010b; Rupérez et al., 2012; Witten, Frank & Hall, 2011), IG has not been explored in biomechanics. Instead of IG, Principal Components Analysis (PCA) has been a popular technique to select/construct features on walking and running biomechanical data (Hoerzer et al., 2015; Maurer et al., 2012; Muniz et al., 2010b; Nigg et al., 2012). Support Vector Machine (SVM) and Artificial Neural Networks (ANN) are techniques that have been used in classification tasks, like in distinguishing effects of velocity (Joo et al., 2014), aging (Begg & Kamruzzaman, 2005; Fukuchi et al., 2011; Wu, Wang & Liu, 2007), gender (Baltich, Maurer & Nigg, 2015; Maurer et al., 2012), diseases (Muniz et al., 2010a; Muniz et al., 2010b; Muniz & Nadal, 2009; Nüesch et al., 2012) and footwear conditions (Trudeau et al., 2015). As a result, age, gender (Maurer et al., 2012; Nigg et al., 2012) and the inter-subject’s movement variability (Federolf, Boyer & Andriacchi, 2013; Von Tscharner, Enders & Maurer, 2013) are intrinsic factors that have shown more influence than shoes characteristics on running biomechanics.

Therefore, when the problem involves assessing the effects of shoes interventions, the feature selection stage has to be carefully conducted since it plays a critical role in minimizing bias and the influence of such intrinsic factors. In this context, even PCA has limitations (Von Tscharner, Enders & Maurer, 2013), and a subject-independent analysis on shoes interventions is still an open issue in the literature. In contrast to PCA, IG is a supervised method that ranks variables individually without applying data transformations, and it has the potential to facilitate the interpretation of the influence of a single variable on the underlying classification task (Begg & Kamruzzaman, 2005; Muniz & Nadal, 2009).

As many studies in the literature (Joo et al., 2014; Oh, Choi & Mun, 2013; Schöllhorn, 2004), we adopted ANN for classification, even though SVM has been also successfully used for the same task (Begg & Kamruzzaman, 2005; Maurer et al., 2012; Nigg et al., 2012; Trudeau et al., 2015). Both SVM and ANN have been producing state-of-the-art results, even though some comparative studies (Begg & Kamruzzaman, 2005; Fischer et al., 2011; Muniz et al., 2010b; Yang et al., 2012) have indicated a slight advantage in favor of SVM. However, SVM were originally designed for binary classification, which involves only two classes (e.g., low versus high resilient), and an effective way of extend it for multiclass classification is a research issue (Hsu & Lin, 2002). On the other hand, an ANN maps straightforwardly into multiclass classification without requiring any further adjustments to approach the problem. In this study, although we have approached shoes resilience and upper characteristics as two binary and independent classification problems, a natural extension of this study will focus on the fact that the effects on running may be related to not only single shoes characteristics, but also to the combination of them, resulting in a classification problem with more than two classes. In this context, ANN represents a more stable scenario for further comparisons between binary and multiclass classification results.

Input variables and feature selection by IG

The 3D joint angular displacement time series was vectorized to a 1,254 dimensional vector (3 lower limb joints × 3 planes of angular displacement ×101 interpolated data points + 3 vectors of ground reaction forces ×101 interpolated data points + 42 discrete calculated kinetic and kinematic features). An input data matrix M was then created (10 trials × 4 shoe condition × 27 subjects), resulting in a matrix dimension of 1,080 × 1,254. The 1,080 lines of M represented each subject trial in terms of 1,254 input variables, some of which may be more affected by specific characteristics of shoe design. In this context, finding a small subset of input variables is a desired result (Rupérez et al., 2012) because that may indicate a more discriminative and less redundant subset of features that would improve the results of the classification.

It was not practical to test all subsets of the 1,254 input variables/columns available, then we used IG to rank the variables in a decreasing order of relevance. As a supervised method, IG ranks an input variable X according to their discriminative power to separate the subject’s trials in terms of a target variable C, like shoes resilience or upper. Usually, a distinc t value ci of a target variable C is known as a class. In our study, C = {low, high} for resilience and C = {minimalist, structured} for upper. IG is a correlation measure based on the information-theoretical concept of entropy, and the entropy of a variable C is defined as in Eq. (1) (Yu & Liu, 2003): (1) HC=−∑iPcilog2Pci

and the entropy of C after observing values of a variable X is defined as Eq. (2) (2) HC|X=−∑jPxj∑iPci|xjlog2Pci|xj

where P(ci) is the prior probabilities for the values of C, and P(ci|xj) is the posterior probabilities of C given the values of X. The amount by which the entropy of C decreases reflects additional information about C provided by X, which is called information gain and it is given by Eq. (3) (3) IGC|X=HC−HC|X.

In our experiments, diverse subsets of variables were tested; they contained an increasing number of variables (25, 50, 100, 150 and 200) and bigger subsets were systematically formed by aggregating less relevant variables according to the IG criteria. This step was crucial to determine the smallest number of variables that achieved accurate discrimination of the resiliencies and upper structures. In this context, the greatest subset of variables involved in our experiments included 200 variables, since our analysis indicated that 200 variables were enough to evaluate the hypotheses of this study.

Classification Procedure

As described previously, lines in the matrix M are instances of subject’s trials, and each trial belongs to a class ci of resilience (low or high) and a class of upper (minimalist or structured). As a supervised classification approach, a subset of subject’s trials was used for training and fitting a classification model, and the rest of the subject’s trials were used for validating the model. To estimate how accurately the classification model will perform in practice, we adopted the standard k-fold-cross-validation, which divides the subject’s trials into k mutually exclusive folds of nearly equal size: k − 1 folds are used for training, and the remaining fold for testing. The procedure repeats k times, such that each fold is used once for validation. For this reason, the validation results are usually averaged over the k rounds. For each training round, a subset of variables was selected by using IG, their values were scaled from −1 to 1, and then an ANN learning model was trained.

The classification accuracy of the resulting ANN model was then computed on the test fold. Because we are dealing with binary classification problems (i.e., two classes of resilience and upper), accuracy is given in terms of the entries of a confusion matrix for positive and negative classes (Han, Pei & Kamber, 2011), as shown in Eq. (4). Given two classes (e.g., low and high resilience), we can express in terms of positive trials (trials of one class, e.g., resilience =low) versus negative trials (e.g., resilience =high). True positives (TP) refer to the positive trials that were correctly classified by the ANN model, while true negatives (TN) are the negative trials that were correctly classified by the ANN model. Similarly, false positives (FP) and false negatives (FN) are the negative and positive trials, respectively, that were incorrectly classified by the ANN model. (4) Accuracy=TP+TNTP+TN+FN+FP.

We used a traditional feed-forward multi-layer perceptron network with 3 layers (Fischer et al., 2011): an input, an output and a single hidden layer (Hastie, Tibshirani & Friedman, 2001). The number of entries in the input layer corresponds to the number of selected variables in the feature selection stage, and the output layer consists of 2 neurons, which corresponds to each class ci in our target variables, i.e., C = {low, high} for resilience or C = {minimalist, structured} for upper. As an ANN requires a parameter setting, which is still a research issue, we perform an exhaustive searching through a subset of parameters values: the number of neurons in the hidden layer was selected from the set {10, 25, 50, 75}; the learning rate ∈{0.05, 0.15, 0.25} and the number of training cycles ∈{300, 700, 1,100, 1, 500}. To reduce the risk of overfitting, we adopted the decay procedure (Hastie, Tibshirani & Friedman, 2001), as implemented in the RapidMiner software.

We conducted the classification in two contexts:

I. a 4-fold-cross-validation for each subject (40 trials) to assess the existence of effects from shoes conditions, finding one accuracy value to discriminate the shoe condition for each subject and for each subset of features. The purpose of considering trials of one subject only is to conclude about our assumptions h1 and h2 in a context that avoids the influence of subject’s intrinsic factors.

II. a standard 10-fold-cross-validation involving all subjects’ trials to assess the existence of subject-independent changes induced by the shoes interventions, finding one accuracy value for each subset of features.

By comparing classification accuracies between the contexts I and II, it was possible to analyze the subject-dependency of the results and provisionally evaluate a pattern induced by different resilience and upper conditions. Classification accuracy of higher than 80% was considered good (Hoerzer et al., 2015), and we reported results when the ANN method achieved the best classification accuracy. The machine learning procedures were conducted in the software RapidMiner (v.5.3.015, Dortmund, Germany).

Results

All 1,254 variables were involved in the experiments of resiliencies and uppers. In both cases, the accuracy of discriminating the effect of shoe conditions indicated that context I outperformed context II.

The composition of the most relevant features subsets according to IG are detailed in Tables 1 and 2 for the midsole resilience and upper structures comparisons, respectively. Although forces have been the most discriminative variables for both midsole and upper, the top five features for upper and resilience came from different components, respectively vertical and medio-lateral forces.

Table 1 Number of variables of each feature subset according to IG rank for midsole resilience comparison.

Resilience comparison	25 most relevant variables	50 most relevant variables	100 most relevant variables	150 most relevant variables	200 most relevant variables	
Medio-lateral GRF	9	11	14	17	18	
Sagittal Ankle	6	9	16	18	25	
Transversal Hip	3	12	17	33	39	
Vertical GRF	2	4	6	12	16	
Antero-posterior GRF	2	2	7	11	17	
Discrete Force Variables	2	3	4	4	4	
Discrete Kinematic Variables	1	1	1	2	2	
Frontal Knee		7	12	12	14	
Sagittal Hip		1	19	35	38	
Transversal Knee			4	5	12	
Frontal Ankle				1	5	
Frontal Hip					6	
Transversal Ankle					4	
% of total features (1,254)	2%	4%	8%	12%	16%	

Table 2 Number of variables of each feature subset according to IG rank for upper structures comparison.

Upper structure comparison	25 most relevant variables	50 most relevant variables	100 most relevant variables	150 most relevant variables	200 most relevant variables	
Antero-posterior GRF	10	19	33	38	40	
Vertical GRF	8	12	16	19	21	
Discrete Force Variables	4	4	4	6	9	
Sagittal Ankle	2	10	20	22	28	
Medio-lateral GRF	1	4	12	18	22	
Transversal Knee		1	7	19	25	
Frontal Ankle			5	13	26	
Transversal Ankle			2	9	10	
Discrete Kinematic Variables			1	2	2	
Sagittal Knee				4	7	
Transversal Hip					6	
Frontal Knee					4	
% of total features (1,254)	2%	4%	8%	12%	16%	

Resilience effect

In context II, 200 variables were sufficient to distinguish midsole resiliencies with an accuracy of 84.8% (Fig. 2, red line). The accuracies indicated that context I, which considers only trials of a single subject, outperformed context II for all subsets of features (Fig. 2, blue line). A mean accuracy of 89.4% (±8.3%) to classify the two resiliencies was reached by considering only the 25 most relevant features, while the best accuracy of 93.9% (±5.0) to classify the two resiliencies was reached with the 150 most relevant features.

Figure 2 Accuracy levels to discriminate midsole resilience materials in various contexts considering different subsets of variables.

Mean accuracy and standard deviation for each subset of input variables with the highest IG values to discriminate resilience materials. Red line represents the context II and considers all subjects together. Blue line represents the context I and considers each subject in isolation.

Among the 200 most relevant variables to discriminate between low and high resilience cushioning materials, six of them were discrete biomechanical variables (4 ground-reaction force variables and 2 kinematic variables) (Fig. 3). The 5 most relevant variables came from medio-lateral force, between 6% and 10% of stance phase.

Figure 3 Ground reaction force and Kinematics time-series during running with different resilience midsoles.

(A) Mean time series of ground reaction force for different resilience of cushioning materials. (B) Mean time series of joints kinematics in all planes of motion for different resilience materials. Blue lines represent the low resilience cushioning condition and red dotted lines represent the high resilience cushioning condition. The 200 highest IG variables are highlighted in the yellow boxes.

Upper structure effect

The results indicated that the upper structures effects were less complex than the cushioning materials ones. In the context II, accuracy higher than 85% was achieved by considering only 25 variables to differentiate upper structures (Fig. 4, red line). As in the case of resiliencies, results on uppers shown that context I outperformed context II; it was possible to obtain a mean accuracy of 93.4% (±4.8%) with 25 variables, and 95.6% (±3.8) with 150 variables (Fig. 4, blue line).

Figure 4 Accuracy levels to discriminate upper strustures in various contexts considering different subsets of variables.

Mean accuracy and standard deviation for each subset of input variables with the highest IG values to classify upper structures. Red line represents the context II and considers all subjects together. Blue line represents the context I and considers each subject in isolation.

Among the 200 most relevant variables to discriminate structured and minimalist uppers 11 of them came from the discrete biomechanical variables (nine force variables and two kinematic variables) (Fig. 5). The five most relevant variables to discriminate between uppers were vertical forces from 11% to 14% of stance phase and the first peak.

Figure 5 Ground reaction force and kinematics time-series during running with different shoe upper structures.

(A) Mean time series of ground reaction force for different shoe upper structures. (B) Mean time series of joints kinematics in all planes of motion for different shoe upper structures. Black lines represent the structured upper condition and Pink dotted lines represent the minimalist upper condition. The 200 highest IG variables are highlighted in the yellow boxes.

Discussion

We proposed an entirely supervised approach based on ANN to distinguish the effects of different midsole resiliencies and upper structures of shoes on running biomechanics. The results confirmed our first and second hypotheses because it was possible to observe the effects on running kinematics and kinetics caused by low and high resilience cushioning midsoles and structured and minimalist uppers by the adopted ML approach. IG was efficient in selecting important features, as was confirmed by the proportionally slower increase in classification accuracy with respect to increasing numbers of input features (Fig. 4). The top biomechanical variables in the IG rank may be considered the most responsible for distinguishing the effects of the upper structures and midsole resiliencies.

When analyzing all subjects together, the methodology successfully differentiated the two resiliencies with 84.8% accuracy using 200 variables and the two shoe uppers with 85.3% accuracy using 25 variables, which is higher than the classification rate threshold of 80% chosen by Hoerzer et al. (2015). Intra-subject analysis increased the classification accuracy for resiliencies to a mean of 89.4% and for uppers using just 25 variables to a mean of 93.4%. This indicated that the adopted ML analysis achieved more accuracy to identify different conditions (cushioning materials and shoe uppers) within a given subject than within the set of all subjects, which is consistent with the higher inter-subject variability.

Among the variables in the interpolated time series, the 25 most relevant features for discriminating midsole resilience were mainly forces, ankle flexion-extension, and hip rotation variables. Sinclair et al. (2014) reported higher tibial acceleration, calcaneus eversion and internal rotation of tibia when running with high resilience midsole shoes. However, due to the different shoe brands, part of these results could be attributed to the midsole resilience, and part of them to other parts of the shoe (differences in the upper, for example). According to our results, the main biomechanical alterations caused by different midsole resilience will be on ground reaction forces, sagittal plane of ankle and transversal plane of hip. Considering the discrete variables that differentiated the resiliencies, four were related to vertical force (first peak and minimal force) and two were related to ankle kinematics (dorsiflexion and eversion). Resilience causes significant effects on the initial impact of the heel with the ground while running. The cushioning materials were inserted only under the heel part of the shoe; considering we focused our study on rearfoot strikers, it was acceptable that the resilience especially influenced the variables related to this first part of the stance phase.

The 200 most relevant features that discriminated the two resiliencies were distributed in short time windows spread over all cycle periods of the kinematic and kinetic time series. This does not mean, however, that these 200 features were equally relevant for all 27 subjects. These short and sparse windows corresponded to individual patterns of biomechanical responses, or a group of individuals with the same biomechanical responses, which leads us to refute part of the third hypothesis and conclude that changes due to shoe midsole resilience seemed to be subject-dependent. This means that is possible to discriminate the two resiliencies with a fewer number of variables and with a higher accuracy if we analyze individualized biomechanical data. In summary, each runner responded differently to resilience in the cushioning materials, changing a different pool of biomechanical variables that represent distinct motor strategies. According to the “muscle tuning paradigm” proposed by Nigg & Wakeling (2001), the individualized ability to modify the muscle tonus in response to impact stimuli is one of the causes of different adaptabilities to shoes observed among subjects. Additionally, the most discriminative kinematic features were not found at the same time periods as the most discriminative kinetic features in the stance phase, suggesting that kinematics adjustments in the lower limbs caused by shoe changes might not be influencing impact attenuation. This differs from what Hennig, Valiant & Liu (1996) and Milani, Hennig & Lafortune (1997) suggested in their studies.

The upper structures classification had higher discriminative power than midsole resiliencies, as was reflected in the higher accuracy levels of upper structure classification. The 25 most relevant features for discriminating upper structures were composed mainly of force variables (both discrete variables and time series features) and sagittal ankle variables. From the 200 features, we found that the most relevant biomechanical variables for the classification of uppers were concentrated in the first and last third of the stance phase for all three-force components, sagittal plane of ankle, and for all planes of knee. The exceptions were for the frontal and transverse planes of the ankle, which had the central third as the most relevant part for classification. Therefore, changes in shoe uppers seem to be more subject-independent.

Eleven discrete biomechanical features were among the most relevant for classifying upper structures. Nine were force variables related to first and second vertical peaks, minimal vertical force, and breaking antero-posterior force; two were from ankle kinematics. According to the relevant discrete parameters, it seemed that force features had highest discriminative power, so were more relevant for differentiating upper structures than kinematic variables and they were temporally distributed across the stance phase. The flexibility of the upper (due to different applied materials) could affect the flexibility of the foot inside the shoe, its own capability to absorb impact by stretching the foot arches and generate the propulsion force, so it is reasonable to assume that the force features across the entire stance phase are relevant variables for distinguishing upper differences. The differences in structure probably affect sensitivity to the ground, and consequently how runners modulate applied force to the ground. As seen by Hagen & Hennig (2009), firmer foot contact with a shoe could result in lower loading rates due to a better coupling between the foot and shoe, which in turn facilitates the use of the impact-absorbing technology of the midsole.

In a training context, it would be ideal if each runner performed a biomechanical assessment of his or her running shoes. Our sample was constituted by rearfoot strikers, which were habituated to classic running shoes (not minimalist). This running strategy is the most prevalent among runners (De Almeida et al., 2015; Hasegawa, Yamauchi & Kraemer, 2007) and according to Lieberman et al. (2010), rearfoot strikers who grew up wearing shoes are more prone to be influenced by shoes conditions, so for this type of runners it is especially important to understand how this external factor (running shoe) changes (or not) each runner’s particular running mechanics. Moreover, only few variables among the 1,254 available (about 12% for different cushioning materials and 4% for different shoe uppers) are evidently necessary to detect alterations of running biomechanics with high accuracy (93.9% ± 5.0% and 94.3% ± 4.5%, respectively). This result is plausible with the theory of “functional groups” (Hoerzer et al., 2015; Nigg et al., 2003) and it is appropriate to understand particular cases. Our results showed that runners have different responses to the materials used in the shoes and a general conclusion arising from a heterogeneous sample may lead to wrong outcomes. Further studies could analyze the influence of smaller sub-groups with similar biomechanical responses to shoe characteristics. This procedure would be useful for understanding how different resilience of cushioning material affects the running mechanics for specific “types” of athletes. It was also possible to demonstrate that a biomechanical study of sports shoes isolating characteristics to be tested provided more specificity in the comprehension of the influence each part on running biomechanics. Future studies could explore other types of feature analysis from biomechanical data such as complexity (Sejdić et al., 2014), or sample entropy (Rathleff et al., 2010; Søndergaard et al., 2010) to distinguish running shoe properties and constructions from the perspective of signal variability.

Conclusion

The applied methodology based on feature selection by IG and classification by ANN successfully differentiated the high and low resilience materials and the structured and minimalist uppers with accuracies of higher than 85% using 200 features (16% of 1,254 available features). The classification of upper structures presented higher accuracy levels than that of midsole resilience probably due to higher inter-individual variability, but in both cases the forces are among the 25 most relevant features subset. The ground reaction forces are the most important features to differentiate midsole resilience and the resilience caused valuable effects on initial heel impact while running. The different patterns of biomechanical response chosen by runners to adapt to different resilience probably led to lower accuracy levels for this classification. We can therefore conclude that biomechanical changes due to shoe midsole resilience seem to be subject-dependent and changes due to upper structure seem to be subject-independent.

We would like to thank Dass Nordeste Calçados e Artigos Esportivos Inc. for supplying the shoes prototypes and the laboratory used in this study, and Alexandro Rodrigo da Rosa and Jonas Schneider for designing and developing the shoes used in this study.

Additional Information and Declarations

Competing Interests

Author Contributions

Human Ethics

Data Availability

The research has been co-funded by Brazilian Agency for Research funding (CAPES) and Dass Nordeste Calçados e Artigos Esportivos S/A. Wagner Oliveira works in Dass Nordeste Calçados e Artigos Esportivos S/A. Andrea N Onodera works in Dass Nordeste Calçados e Artigos Esportivos S/A and conducts her PhD supervised by Dr Isabel CN Sacco. Mrs Maria Isabel Roveri is part of the primary authors PhD, but is not associated to Dass Nordeste Calçados e Artigos Esportivos S/A and is funded by a CAPES scholarship. Data was collected, analyzed and the paper written with no influence from the funding agencies or shoe company, and no author will receive anything of value from the commercial products included in this paper.

The authors, therefore, have no competing interests.

Andrea N. Onodera conceived and designed the experiments, performed the experiments, analyzed the data, contributed reagents/materials/analysis tools, wrote the paper, prepared figures and/or tables, reviewed drafts of the paper.

Wilson P. Gavião Neto conceived and designed the experiments, analyzed the data, contributed reagents/materials/analysis tools, wrote the paper, prepared figures and/or tables, reviewed drafts of the paper.

Maria Isabel Roveri and Wagner R. Oliveira conceived and designed the experiments, performed the experiments, analyzed the data, wrote the paper, reviewed drafts of the paper.

Isabel CN Sacco conceived and designed the experiments, analyzed the data, contributed reagents/materials/analysis tools, wrote the paper, reviewed drafts of the paper.

The following information was supplied relating to ethical approvals (i.e., approving body and any reference numbers):

The School of Medicine at the University of Sao Paulo granted Ethical approval to carry out this study (Ethical Application CEP-FMUSP: Protocol #054/14).

The following information was supplied regarding data availability:

Figshare: https://figshare.com/s/6530932d478b54ff5d6c.

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
