# Peer review of "Immediate effects of EVA midsole resilience and upper shoe structure on running biomechanics: a machine learning approach"

_PeerJ, doi:10.7717/peerj.3026_

## Round 0.1 · original submission · Major Revisions

Although all the comments were brief, two out of three of the reviewers recommended Major Revision. Please revise your paper based on the comments.

Reviewer 1 ·

Basic reporting

The manuscript is well-written and the information is properly presented and structured in accordance with the journal requirements.

Experimental design

The information provided focuses on a research which is within the scope of the journal.

Validity of the findings

Novelty of the research could be improved in the introduction section.
The introduction should include some studies (references), if available, on EVA-based applications on midsole shoe structure.
The study is conducted using EVA as a base material, which has not been characterized throughout the text, and, in my opinion, some research focused on this material should have been carried out in order to obtain better and complete conclusions.

Additional comments

Some relations between EVA properties and the biomechanical properties of the shoe should be addressed.

·

Basic reporting

No Comments

Experimental design

No Comments

Validity of the findings

No Comments

Additional comments

This paper conducted a machine learning analysis to identify features. It is interesting.

(1) How do you enroll the subjects?
(2) Why do you filter the marker coordinates using 12 Hz zero-lag 4th-order low-pass Butterworth filter?
(3) Please give the mathematical expression of butterworth filter
(4) Line 208, please define IG.
(5) Why support vector machine is not used? It is also a good classifier. But the author did not mention it at all.
(6) Line 255, what is the incremental step for {10,…,75};
(7) How do you apply ANN in this study, please explain it clearly.
(8) What is the structure of ANN?
(9) Some ANN-related papers are not seen in this paper, for example,
a. Preetha Phillips. Fruit Classification by Biogeography-based Optimization and Feedforward Neural Network. Expert Systems. 2016, 33(3): 239-253
b. Zhihai Lu. A Pathological Brain Detection System Based on Radial Basis Function Neural Network. Journal of Medical Imaging and Health Informatics. 2016, 6(5): 1218-1222

Reviewer 3 ·

Basic reporting

- The manuscript adheres to PeerJ templates, but parts of it are very confusing to understand.

Experimental design

- Very confusing. Even after reading it several times, I am still not sure what you actually classified. Please rewrite this section to make more clear.

Validity of the findings

- I don't understand what are those reported numbers in Figure 2 and Figure 4. It is probably due to a lack of details in the methodology section, but they are difficult to understand.
- You should discuss some of recent gait papers (e.g., https://www.ncbi.nlm.nih.gov/pubmed/23751971) and how those features can be related to some of you used in your own work.

Additional comments

- It's not clear from the manuscript why these results would matter to a general science audience. It seems more suitable for a shoe manufacturer. Hence, the authors should emphasize the science part.

---

## Round 0.2 · accepted · Accept

The authors have addressed all the comments raised in the previous review round.